# “The Heart Asks Pleasure First”—Conceptualizing Psychiatric Diseases as *MAINTENANCE* Network Dysfunctions through Insights from slMFB DBS in Depression and Obsessive–Compulsive Disorder

**DOI:** 10.3390/brainsci12040438

**Published:** 2022-03-25

**Authors:** Volker A. Coenen, Thomas E. Schlaepfer, Bastian E. A. Sajonz, Peter C. Reinacher, Máté D. Döbrössy, Marco Reisert

**Affiliations:** 1Department of Stereotactic and Functional Neurosurgery, Medical Center of Freiburg University, 79106 Freiburg, Germany; bastian.sajonz@uniklinik-freiburg.de (B.E.A.S.); peter.reinacher@uniklinik-freiburg.de (P.C.R.); mate.dobrossy@uniklinik-freiburg.de (M.D.D.); marco.reisert@uniklinik-freiburg.de (M.R.); 2Medical Faculty, Freiburg University, 79106 Freiburg, Germany; thomas.schlaepfer@uniklinik-freiburg.de; 3Center for Deep Brain Stimulation, Medical Center of Freiburg University, 79106 Freiburg, Germany; 4Laboratory of Stereotaxy and Interventional Neurosciences, Department of Stereotactic and Functional, Neurosurgery, Medical Center of Freiburg University, 79106 Freiburg, Germany; 5Department of Interventional Biological Psychiatry, Medical Center of University of Freiburg, 79106 Freiburg, Germany; 6Fraunhofer Institute for Laser Technology (ILT), 52074 Aachen, Germany; 7Faculty of Biology, University of Freiburg, 79104 Freiburg, Germany; 8Department of Diagnostic and Interventional Radiology, Medical Physics, Medical Center of University of Freiburg, 79106 Freiburg, Germany

**Keywords:** brain networks, dopamine, deep brain stimulation, MAINTENANCE, major depression, OCD, reward system, SEEKING, treatment resistant depression, TRD

## Abstract

More than a decade ago, deep brain stimulation (DBS) of the superolateral medial forebrain bundle (slMFB), as part of the greater MFB system, had been proposed as a putative yet experimental treatment strategy for therapy refractory depression (TRD) and later for obsessive–compulsive disorders (OCD). Antidepressant and anti-OCD efficacy have been shown in open case series and smaller trials and were independently replicated. The MFB is anato-physiologically confluent with the SEEKING system promoting euphoric drive, reward anticipation and reward; functions realized through the mesocorticolimbic dopaminergic system. Growing clinical experience concerning surgical and stimulation aspects from a larger number of patients shows an MFB functionality beyond SEEKING and now re-informs the scientific rationale concerning the MFB’s (patho-) physiology. In this white paper, we combine observations from more than 75 cases of slMFB DBS. We integrate these observations with a selected literature review to provide a new neuroethological view on the MFB. We here formulate a re-interpretation of the MFB as the main structure of an integrated SEEKING/MAINTENANCE circuitry, allowing for individual homeostasis and well-being through emotional arousal, basic and higher affect valence, bodily reactions, motor programing, vigor and flexible behavior, as the basis for the antidepressant and anti-OCD efficacy.

The Heart asks Pleasure first,And then Excuse from Pain,And then those little Anodynes,That deaden suffering;  And then to go to sleepAnd then, if it should beThe will of its InquisitorThe liberty to die.—Emily Dickinson (1830–1886)

## 1. Introduction

Emotions—psychological states with biological substrates—are vital for the initiation and pursuit of flexible behavior. Despite the subjective experiences, or feelings, associated with them, it appears to be justified to regard distinct emotions as the brain’s information classifiers, allowing adaptive behavior in an environment which confronts the organism with a great complexity of parallel interoceptive and exteroceptive signals for the animal’s homeostasis. Therapy refractory depression (TRD) and obsessive–compulsive disorder (OCD) are characterized by deficiencies in showing flexible behavior and one reason is suspected to be a deficiency in the automated classification of emotion related information (affect valence) [1].

Deep Brain Stimulation (DBS) of the superolateral branch of the medial forebrain bundle (slMFB) is a therapy under investigation for TRD and OCD [2,3,4]. Hypomania in subthalamic nucleus (STN) DBS, as a side effect of high frequency stimulation in Parkinson’s disease [5], has been attributed to a pathological co-activation of the reward or SEEKING system [6]. Panksepp’s SEEKING system (capitalization refers to an anatomical system responding to intracranial electrical stimulation) entails the idea that it is not only the consummatory phase of reward but the animal’s euphoric drive which makes the organism explore (seek) for a reward, which in turn characterizes this system [7,8,9]. In our current understanding, the biological substrate of SEEKING is the medial forebrain bundle (MFB) and this network can be accessed in its midbrain terminal field—located just medial and outside the STN (the greater MFB system), a region called the lateral ventral tegmental area (VTA), subserved by feedback fibers from the prefrontal cortex (PFC) as the slMFB (superolateral medial forebrain bundle) [10]. DBS of this region (slMFB close to the VTA) was consequently introduced as a therapy under investigation for TRD [2]. Since this time, at our institution, more than 75 patients have been implanted, mainly in the indications of TRD and obsessive–compulsive disorder (OCD). More than 30 treated patients have been reported so far in the literature [2,3,4,11,12], mostly suffering from TRD and often with favorable outcomes [13,14,15]. 

Since a part of the MFB system (the imMFB) is, moreover, confluent with the main structure of the mesolimbic and mesocortical dopaminergic system, the whole system (MFB) has been understood as a network mediating reward, reward anticipation and motivation and, therefore, shows strong antidepressant efficacy upon modulation [2,6,15,16,17,18,19]. Other transmitters such as Glutamate (GLU), Serotonin (SE), Orexin (Orex) and others play important roles. Part of the stimulation effect has been explained with the activation of descending Glu fibers (as part of the slMFB and utilizing it as the feedback loop it is) which activates the VTA DA neurons and the midline imMFB with its DA projection to the PFC [14,20,21]. With growing experience in the clinical management of patients and with an enlarging body of evidence from preclinical studies (for a review see [14,17]), a slightly different interpretation of the physiological and pathophysiological function of the target structure emerges. Accumulating data suggest an additional regulator role of the MFB with respect to a multiplicity of functions, complementary but beyond mere reward or SEEKING. This regulator role had already been expected and discussed, given the far-reaching subcortical and cortical connectivity of the slMFB [6,18,19,22]. We postulate that as such, the MFB as an integration belongs to the SEEKING but also to an overarching MAINTENANCE system (Figure 1). Furthermore, we are certain that DA plays an important regulating role beyond signaling reward and aversiveness [23,24]. Insofar, the definition of such a MAINTENANCE system is strongly associated with a new role given to the transmitter DA.

In this perspective paper we perform a selective review of the pertinent published data on the slMFB and will attempt a new interpretation of its role, in the light of anecdotal clinical data from observations around slMFB DBS surgeries. We will extrapolate towards a re-interpretation of the slMFB as the main structure of a here newly defined MAINTENANCE circuitry, which governs the valence of basal emotional states (aversive vs. appetitive) and serves to drive emotional arousal, basic and higher affect valence, bodily reactions, motor programs, vigor and flexible behavior, accordingly. Especially in the therapy resistant diseases treated with slMFB DBS, a dysfunction also related to the wider Dopamine (DA) system is contingent. Consequently, there is reason to interpret psychiatric diseases like OCD and TRD as network dysfunctions of the MFB and, therefore, of the MAINTENANCE system.

## 2. Methods

We performed a literature review regarding the basal ganglia and their integration with the MFB/slMFB, slMFB (anatomy, physiology, pathophysiology)—networks relevant for depression and OCD—and slMFB DBS (in TRD and OCD). We selected the work which helps to substantiate our view on the human MFB as a MAINTENANCE pathway. We added anecdotal information from our experience concerning intra- and perioperative physiological and behavioral observations in n = 75 patients from open case series (OCD) and the FORESEE, FORESEE II and FORESEE III trials (all in treatment resistant depression, TRD). We have in the following interpreted this information toward a system conceptualization regarding the slMFB, with a special focus on the role of DA. This approach appears to be justified for a perspective paper.

## 3. Results

### 3.1. Literature Review

The identified literature is mentioned in the context of the discussion and will be found under the distinct sub-headings.

### 3.2. Anecdotal Observations from DBS Surgery

Test stimulation, emotional responses: A typical response to the intraoperative test stimulation (which is usually performed unilaterally in the center of the slMFB with 130 Hz, 60 μs, 1–2.5 mA) is appetitive motivation. Patients—after a reduction in sedation when prompted to respond to 5–10 min of stimulation—are clearly more awake and attentive (behavioral arousal) beyond the pre-sedation level. They sometimes report the emergence of motivational ideas (e.g., “I would like to be under the Christmas tree with my kids”; “I would like to go on holiday with my spouse” etc.). Stimulation responses on the OR table are rarely pleasant beyond the abovementioned motivational response [25]. Conversely, another response to stimulation could be: “I do not like this” and this is typically associated with a perception of the patient’s heart beating (palpitations), usually due to a heart rate increase. When asked, patients report an effect akin to having consumed too much caffeine. This effect wears off immediately after test stimulation is stopped and is rarely (if at all) seen during chronic stimulation.

Test stimulation, autonomic responses: At the optimal stimulation point, anesthesiologists typically perceive an increase in the heart rate and the blood pressure. This is a transient phenomenon which wears off under 2–3 min of constant stimulation [10,25] in all patients who are not under beta-receptor blocking medication.

Test stimulation, oculomotor responses: Higher stimulation amplitudes elicit an activation of ipsilateral oculomotor nerve fibers and typically an adduction and down forced movement of the ipsilateral eye. At the lower border of the slMFB, this effect is used to guide the implantation depth and should occur at an amplitude of 1.5–2 mA. We have seen this effect in every patient.

DBS electrode placement and micro-lesioning effects: All patients with more agitated forms of TRD, and also with OCD, upon implantation of the permanent DBS electrode have typically reported “a relief like taking away a weight from the chest”. In OCD, this might typically be the point when obsessions are drastically diminished. Patients are typically more calm (less behavioral arousal) and show a much more normalized (decreased) heart rate. Postoperative effects: Patients after bilateral slMFB DBS electrode implantation have reported improved mood, motivation, decreased anxiety and occasionally show somewhat reduced activity. It is not clear if this an after-echo of the intraoperative stimulation or a clear lesioning effect. With the use of the Boston Scientific Cartesia (segmented) lead (Boston Scientific, Valencia, CA, USA), we have observed some 2–3 days delayed micro-lesioning effect (similar to observations in movement disorder surgeries). We have seen some instances where patients showed dramatically reduced behavioral arousal in the postoperative period (1 week), associated with some increased need for sleep. Imaging workup (cranial computed tomography) did not show any signs of bleeding and the effect completely resolved spontaneously after a maximum of ten days. During this period, patients reported less obsessions and anxiety but were also less attentive.

## 4. Discussion

Emotions are vital for the organization for the initiation of movements and for pursuing flexible behavior. Affect is the immediate (cross-sectional) form of emotion and is accompanied by a congruent subjective experience. Affect valence of exteroceptive or interoceptive brain signals can be conceptualized as a process of coarse information categorization for a plethora of primary information relevant for social interaction. The process of affect valence allows an organism (e.g., primates, including the human primate) to integrate dynamic and contextual information and, as a consequence, to flexibly move, behave and interact in a social context [26]. The categorization into emotions needs to happen fast and is primarily performed unconsciously (although in its consequence an emotion can be consciously perceived). Evolutionarily, different higher emotions are realized in distinct brain systems. Panksepp has described four (later, seven) emotional systems: SEEKING, LUST, CARE, PLAY, PANIC/GRIEF, FEAR, RAGE, which he derived from animal experiments. We will not touch in detail upon all of these systems but merely acknowledge their existence [27]. 

DBS of the medial forebrain bundle is essentially a stimulation of the mesencephalic ventral tegmentum (MVT) targeting the descending slMFB fibers, which enter the VTA and terminate there. This surgical strategy has been pursued for more than a decade [2,3,11,12,28,29] and the implantation technique, including the acute effects, has been thoroughly described [25,30]. For long term stimulation effects, we refer to the pertinent publications [3,31]. In addition to diffusion tensor magnetic resonance imaging fiber tractography (DTI FT) based targeting, intraoperative microelectrode-recording together with acute test stimulation have been the standard of implantation in most reported case series [2,3,11,12]. The inadequate utilization of these techniques has in some reported cases led to diminished antidepressant efficacy [28].

### 4.1. Basal Ganglia, the (Human) MFB and the Role of DA

The importance of the basal ganglia as part of the extrapyramidal motor system is clear but is, in the details, not entirely cleared up. Although their role in motor and behavioral control is undisputed, an integration with the recently introduced human MFB concept [18,19,22] is missing. Extrapyramidal control of movement and behavior integrates motivational states. It uses complex regulation mechanisms, which include the basal ganglia and are not yet fully understood [32,33,34]. The basal ganglia are functionally integrated with the frontal cortex [35]. In early descriptions, the basal ganglia participate in five cortico-striato-thalamo-cortical parallel regulation circuits: motor, oculomotor, dorsolateral prefrontal, lateral orbitofrontal and anterior cingulate cortex circuits [35]. Disturbances of these parallel loops have been utilized to explain neuropsychiatric disorders (especially OCD) [36]. This view is generally endorsed in pertinent reviews, although the number of loops is brought down to three: associative, motor and limbic [37,38]. Another functional segregation within these loops is the indirect, direct, and hyperdirect pathway concept (Figure 1) [32,39]. The direct pathway is largely associated with D1 receptors and has a role in facilitating thalamocortical disinhibition, which in turn releases cortical motor activity. The indirect pathway functions via a D2 mechanism and inhibits thalamocortical activity leading to an inhibition of movements [39]. In contrast to earlier views, the role of indirect and direct pathways is today understood as a finely balanced motor control system with parallel activation during movement performance. The direct pathway in this context, facilitates planned motivated movements, while the indirect pathway is thought to suppress non directly planned movements of inferior motivation [40]. The hyperdirect pathway has already been described decades ago [41,42], but its functional significance has only recently been understood [38,39,43,44,45,46,47]. As a monosynaptic glutamatergic projection, the hyperdirect pathway has a direct connection to the subthalamic nucleus (STN), which is understood as an additional access to the basal ganglia [39] (Figure 2). As such, the STN has an important role in delaying decisions and emitting stop signals in order to allow for a flexible motor program design [48]. Via the hyperdirect pathway, the STN is under the command of the lateral OFC (orbitofrontal cortex), which has an important role in response inhibition [49,50]. In its simplest interpretation, taking the STN (or the hyperdirect pathway) out of the basal ganglia circuitry leads to impulsivity [48]. 

The described pathways (direct, indirect and hyperdirect) work in a complex interaction and participate in the regulation of flexible (motor) behavior. The slMFB integrates with the basal ganglia at strategic connection points (PFC, NAC) and potentially at the VTA/STN [10,51]. The MFB signals distinct levels of motivation (appetence <> aversiveness) and enters the BG circuit with a DA projection to the NAC. The NAC itself modulates the internal globus pallidus (GPi) via a dominantly GABA projection and influences the thalamo-cortical transmission.

### 4.2. Brain Wide Networks Relevant for Depression and OCD

Distinct emotional systems—albeit in a diminutive form—already exist in non-human species and are in part characterized by their activation and behavioral read outs through electrical stimulation [27,52]. These systems can be extrapolated towards a much more complex human anatomy and researchers have described these structures as networks. For a more detailed discussion we refer to [19]. Owing to the distinct symptoms of depression (and OCD) it is possible to describe four basic interacting networks [53]. These networks are reward, affect, default mode and cognitive control. In previous work, we have extended these cortically located networks to subcortical fiber anatomy [19]. In this contribution, we add an overarching system for homeostasis of the organism, the MAINTENANCE system (Figure 3).

In order to better understand the interplay of these networks it might be important to highlight certain anatomical substrates of emotion, as part of these organizational structures. When looking at the anatomical structures involved in the mentioned networks [53] it becomes clear that aside from the VTA, which is somewhat a focus of this work, the axis amygdala to the OFC (orbitofrontal cortex) potentially needs some emphasis. The amygdala connects via distinct pathways to the OFC and one of them is certainly via slMFB/MAINTENANCE (Figure 3 and Figure 4). The multiple connections of the amygdala and the prefrontal cortex (PFC) have been investigated in humans with diffusion-weighted imaging (DWI) [55] and have also been described with respect to the VTA’s bilateral connectivity in non-human species [56]. Recently, Sorovia et al. found a reduced amygdala connectivity in later alcohol use disorder with a history of childhood trauma [57]. Subcortical projection pathways appear to show a convergence of fibers from the ventricular thalamus (mediodorsal thalamus, MDT) which belong to the affect network with the OFC. The same OFC convergence region has been shown for the MFB [19]. In this network interplay, the OFC obviously has an important function. Distinct parts of the OFC play different roles in the top-down control. The lateral OFC has a role in evaluating punishers, while the medial and central OFC evaluate reward and are implicated in reward learning. Such functional distinction has implications also for the subcortical wiring connections between the VTA, subthalamic nucleus and OFC [10].

Repetitive transcranial magnetic stimulation (rTMS) is a non-invasive stimulation technique which is used in TRD [58] and OCD [59,60] and has in part shown promising results. Aside from a discussion about efficacy, TMS is interesting since its application helps our understanding of the function of the regarded networks, especially in distinct network hubs reachable with rTMS. It is of interest in this context to discuss how learning memories in the context of fear will gauge the system to respond to future threats. To create an adaptive fear response the brain must develop sensory cues and contextual memory to understand an aversive situation [61]. The regions which are involved are the amygdala, hippocampus and the ventromedial PFC (vmPFC), Brodmann’s area (BA) 24 and the orbitofrontal region (BA25 and BA32). Fear learning is impeded in bilateral amygdala damage, while declarative memory is impeded with hippocampal damage, leaving fear learning unimpeded. In this context it is of interest to show that fear conditioning can be modulated through rTMS in the dorsolateral PFC, as such when during the application of a fear reminder stimulation is performed [62]. This application hints towards a role of the interplay between top-down and bottom-up emotional responses and to a role of the dlPFC in fear learning and reconsolidation. 

In our understanding, the vmPFC and OFC take special roles in the interpretation of exteroceptive and interoceptive stimuli and, moreover, in the emotional valence process (Figure 5). The vmPFC/OFC complex serves important roles, e.g., in fear conditioning (the PANIC system [27]). Anterior parts of the vmPFC are more active during safety signals, while more posterior parts interpret fear/anxiety related contexts. It appears that the OFC serves as an integrator of the top-down appraisal of an emotional context from the dlPFC and the bottom-up emotional valence. For a recent review in this context see [63].

### 4.3. The Human Medial Forebrain Bundle (MFB)

Most of the information concerning the mfb is derived from rodent anatomy [64,65,66,67,68]. The mfb follows the much simpler anatomy of the rodent brain in which the VTA, the lateral hypothalamus, the NAC and the olfactory bulb perform a much simpler line up. In the workup of a case of hypomania during deep brain stimulation (DBS) of the subthalamic nucleus in the Parkinson’s indication, the human MFB was first described [5], including the primate typical superolateral branch (slMFB), which was in the following further characterized [6,18,22] and recently described as a projection paralleling the hyperdirect pathway, and as such, constituting a feedback pathway from the PFC to the VTA [10] (Figure 6). In our understanding, the slMFB (superolateral MFB) is part of a greater (MFB) system [10] which comprises connections of the VTA to many parts of the brain. Previous descriptions include the imMFB (inferomedial MFB) and the slMFB as well as the motor MFB [22,69]. It is important to understand the MFB as a circuit system in the primate brain. The imMFB is in our understanding essentially confluent with the mesocorticolimbic dopaminergic pathways and as such, realizes the modulation of subcortical and cortical (PFC) structures mediating reward and reward learning, as well as aversive signaling. It is the structure most reminiscent of the rodent mfb [64,66]. The slMFB in turn is in part a Glu pathway which fulfills the criteria of a feedback loop to the VTA as has previously been anticipated [10,56,70,71] (Figure 2 and Figure 3). It is our interpretation that parts of this laterally located slMFB must be bidirectionally connected to the PFC. The slMFB’s portion between the VTA and NAC (Figure 2 and Figure 3) potentially describes an additional and further DA projection to the superior part of the NAC [72], which takes the course of the slMFB. Thus, DA dominantly follows the imMFB to the inferior NAC and distinct regions of the PFC for innervation [73], while a second superior projection (as part of the slMFB) potentially reaches the superior part of the NAC [22].

### 4.4. Clinical Responses to Stimulation and Implantation (slMFB DBS) and the Role of Dopamine

Acute intraoperative high frequency stimulation of the slMFB [25] has effects which can in part be attributed to the DA system, especially the appetitive motivation response [3,14,16,17,74,75]. Other effects shed further light on the role of the MFB for the organism’s homeostasis and allow for a further interpretation towards its role in the physiological regulation of behavior leading to the definition of a MAINTENANCE system. 

The first and potentially most intriguing response to acute unilateral slMFB stimulation is the autonomic one (arousal, heart rate, blood pressure, respiration). We have interpreted this response on the one hand with an increase in DA and Noradrenaline (NA) transmission within the MFB, as has been described in the classical work [76,77]. However, the VTA itself is one of the secondary arousal systems to the ARAS (ascending reticular activating system) [78]. The DA groups A10, 9, and 8 contribute to reward-related behavioral arousal and can—as in our case—be activated through stimulation of descending Glu fibers which enter the VTA [14,20]. This mechanism equips the system with an arousal that prepares the bodily system for a future or parallel behavioral response [71,79]) (Figure 7). It therefore comprises an ideal synergy between the proposed MAINTENANCE system and its integrated SEEKING system (Figure 5). On the contrary, we have seen effects in a minority of our patients which reminded us of selective lesions of the DA cell groups A10 and A9, which have been described in the literature [80]. In these patients (both TRD and OCD), the mere introduction of the DBS electrodes—obviously inducing a micro-lesioning effect—appeared to silence a particular arousal system. The patients showed reduced behavioral arousal and attention and some hypersomnia [78]. This behavior was reversed spontaneously after some days without any sequelae. With reduced arousal and reduced conscious focus, obsessions and agitation were clearly dampened only to reoccur after the resolution of this effect. It is a known fact that lesions to the VTA DA cell groups result in all sorts of frontal lobe behavior [81]. These focal and transient lesions for the first time are reported in humans. A lesion of these VTA DA components to a much lesser extent (without a reduced arousal) led to decreased anxiety. This also occurred upon intraoperative stimulation and the consecutive insertion of DBS electrodes. We interpret this effect with reduced firing of distinct VTA DA neuron populations which show pathologically high activity [82], potentially as a disease consequence. Chronic stimulation of such DA systems might have an anti-OCD effect [4] because DBS feeds a distinct frequency spectrum into this system. As a mechanism, we propose the alteration of a high frequency bursting state which is perceived as an “anxiety signal” by the PFC and is the result of social defeat stress [83].

### 4.5. A MAINTENANCE System and Its Proposed Functions

The MFB is an ancient and evolutionary conserved system that promotes appetitive drive, already realized in much older species [9,27,52], and acts upon electrical stimulation [27,85,86,87]. This system is present in all vertebrates and constitutes a massive close to midline fiber structure in the primate brain. Recent clinical work from self-face experiments [88] points to a more important function in demonstrating a strong activation when (subliminally) confronting the system (VTA) with self portraits. A mere hedonic response to the self face does not appear to be the main effect. The VTA (and therewith the MFB) appears to subserve integrated affective modes (SEEKING and MAINTENANCE) and in part, there is a functional–anatomical integration and no segregation into modules [89]. Immediate responses to unilateral intraoperative high frequency stimulation (slMFB DBS) in TRD and OCD show further physiological and emotional responses, which in our interpretation, are not linked to a pure SEEKING response. In this respect, the MFB incorporates the SEEKING system (for reward and its anticipation, imMFB) and, with the feedback via the dlPFC and OFC (slMFB), realizes the MAINTENANCE functionality by extending and building upon the SEEKING anatomy (Figure 1 and Figure 3). In our view the MAINTENANCE system is responsible for an organism‘s homeostasis and evolution and appears to have linked reward and reward associated behavior (SEEKING) with the realization of this function. The transmitter DA has been typically associated with motivational drive and the motor response to it [24]. Aside from its function in reward anticipation, DA thus might have a regulatory role, allowing for a basic affect valence into appetitive or aversive (neutral), and such signaling is dependent on VTA dopamine neuron activity.

Key concepts of the MAINTENANCE system (Figure 7):In the context of this work, emotions are interpreted as information classifiers, allowing the rapid interpretation of complex interoceptive and exteroceptive stimuli: affect valence.The classical SEEKING system can be integrated into a homeostasis network, the MAINTENANCE system (Figure 5 and Figure 7).MAINTENANCE serves to interpret the most basic “affective modes”: **appetence** vs. **aversiveness** vs. **neutrality (bluntness)**.As such, the greater MFB does not comprise a “module” but acts in different modes since it valences aversive vs. neutral vs. appetitive signaling [89].MAINTENANCE equally interprets appetitive and aversive interoceptive and exteroceptive signals (also with a projection of anticipated consequences in the future) and equips these signals with a very basal “emotional color” (do engage, do not engage).In humans, and a highly developed prefrontal cortex, this interpretation needs to be reconciled with memorized experience, indicating an involvement of the Papez circuit and, therefore, of the mammillary bodies (VTA–MB complex) and interconnection at distinct evolutionary and anatomical levels (Figure 4 and Figure 7).Emotional signaling and the interpretation of this signal in the *MAINTENANCE* system leads to motor responses (engage ⇆ stop ⇆ withdraw) if the top-down emotion appraisal allows (Figure 6).The MAINTENANCE system induces an overall and behavioral arousal andsets autonomous bodily conditions to allow arousal accompanied by a motor response (heart rate, blood pressure, breathing). These settings are performed subconsciously and even before an emotional feeling is perceived. Price, in this respect, speaks from the “visceromotor network” [90].In a further **higher emotional valence**, distinct emotions are categorized (e.g., Panksepp’s seven emotional systems).Depending on the result of this higher valence, a specific motor program is executed.At each step of the evaluation, a top-down emotion appraisal (Figure 6) can be performed in cases where conscious focus is brought to the process.

Consequences for MD and OCD:
OCD: intrusive thoughts (from dlPFC) enter the MAINTENANCE system (VTA) and are attributed as aversive (ego-dystonic). A resulting motor program is then phenotypically expressed as (meaningless) compulsive action. High activity (Figure 7, OCD).
Comorbid depression? The potential overload of the MAINTENANCE system, which only deals with the interpretation of obsessions, and therewith, blocks behavior which serves the organisms’ homeostasis.TRD: a dysfunction of the MAINTENANCE system with a loss of interoceptive and exteroceptive signal evaluation (Figure 7, TRD).
Degradation.Potentially a (neurodegenerative?) loss of Dopamine transmission out of the VTA. As a result, no involuntary (bottom-up) motor program is formed, which leads to reduced overall activity.With increasing dysfunction in bottom-up emotion valence, decision making for motor programing has to be increasingly solved via the top-down system,which in turn occupies part of the cognitive reserve. Increased cognitive decision making thus is exhausting and might lead to decreased cognitive performance.

## 5. Future Perspective for slMFB DBS

With the results of the studies that have been conducted so far, an avenue to future research might be envisioned. DBS of the slMFB, and its proposed mechanism of action, highlights the role of DA and the greater reward/MAINTENANCE system in depression and OCD genesis, albeit our ideas for pathology are distinct for both pathologies (see above). It is one focus of our research to look at monoaminergic and other transmitters’ transmissions in a translational approach [17,20,75,91,92,93,94]. The region of stimulation—the mesencephalic ventral tegmentum—as the seat of the A10 DA cellular group in conjunction with the mammillary bodies, forms a complex basal regulation center, relevant as a basis for the MAINTENANCE system but also for positive affect display [95]. 

At this moment, there appears to be a lack of clinical infrastructure of (biological) psychiatric units which can deal with DBS in TRD and OCD [96], potentially due to an inert apprehension of general psychiatry to pursue such an avenue of treatment. At the same time, there is big interest amongst neurosurgeons to promote these therapies [97]. The guidelines for DBS in psychiatric indications at this moment, reserve this treatment for absolutely refractory cases (see selection criteria in [2,12]) and there are many ethical implications which are far beyond the scope of this work. For a review, please refer to [98]. Despite the expected effectiveness of slMFB DBS in both indications, much larger patient numbers are needed to prove efficacy for this type of research, moving it in the direction of a clinical effectiveness which allows funding through insurances or the public hand. Khan et al. estimate trials as large as n > 100 patients to be statistically valid [99]. These assumptions are mostly based on trials in medication-naive depression. Large trials dealing with DBS in treatment resistant psychiatric implications were not able to show DBS efficacy [100,101] and have been shown to suffer from idiosyncrasies such as too short observation times, variations in surgical targeting and patient selection. We have, therefore, designed a previous gateway trial [12] in order to circumvent some of these problems in our currently recruiting randomized controlled trial (ClinicalTrials.gov Identifier: NCT03653858) for TRD. Moreover, with the growing understanding of the substrate we gain more understanding of disease mechanisms.

## 6. Limitations

It is conceivable to include many of the MAINTENANCE system’s functions into the reward circuitry (SEEKING) itself. This argumentative route was followed by Wise et al. [87,102,103,104]. The interpretation of acute intraoperative stimulation effects is difficult since the patient is typically in a changing state of awake anesthesia, only allowing for a gross neurological examination and for an appreciation of the patient’s current emotional states. However, the interpretation of the latter might be heavily influenced by the systemic float off of drugs like Remifentanyl (Ultiva™, GlaxoSmithKline, Brentford, UK). Ideally, the reporting of acute stimulation effects would be less anecdotal.

## 7. Conclusions

In this perspective paper, we have extended our previous view on the function and role of the MFB for reward and SEEKING by integrating recent insights from slMFB DBS in TRD and OCD. We discussed the MFB as a clearly integrated system with the basal ganglia function and were able to extend this view into a network perspective, including some exemplary discussion of fear memory generation. It is important to understand that behavior is regulated by appetitive drives or aversive punishers [89]. As an extension of its hitherto defined function as the SEEKING system, we allocated a further extended role to the MFB, that is the brain‘s homeostasis or MAINTENANCE system. This theory extends previous views on the MFB. Realizing an organism’s homeostasis through appetitive drive appears to be a totally naturalistic view on the steering of flexible behavior. An organism is more likely to engage towards an appetitive drive [105] than to avoid an aversive one (hence: “the heart asks pleasure, first”), but at least the human primate‘s brain is under the dominating control of the top-down dlPFC system. Our organism, in the context of a homeostatic system, is driven by the results of emotional valence (bottom-up, “emotional color”) which are moderated by cognitive appraisal (top-down, utilizing cognitive reserve). A higher integration of this information is probably achieved in the vmPFC/OFC. As a result, the MAINTENANCE system generates motor programs. They are the substrate of flexible behavior. These functions can be executed by the brain‘s integrated SEEKING/MAINTENANCE systems, utilizing dopamine as its central transmitter in the context of prefrontal executive function. Translational approaches show that a stimulation in the region of the VTA is activating and rewarding and that these signals are DA related [87,102]. However, DA mechanisms are not the only mechanisms we suspect [17,91,106]. Certainly, there must be non-DA and also antidromic stimulation effects involved. 

The MAINTENANCE system overarches the SEEKING system and therewith becomes a regulator of the latter. This role is dependent on a functioning mesocorticolimbic dopamine system and as such in its feedback control, structurally. It is, therefore, dependent on the slMFB and its far reaching subcortical and cortical reward related connectivity. We would in this context like to emphasize once more that the human MFB is a circuit system composed of subcomponents (slMFB, imMFB and motorMFB), which in its entirety constitutes the MAINTENANCE system. Stimulating descending fibers of the slMFB at the position of the MVT (DBS) clearly has activating orthodromic effects on the VTA. The slMFB is not confluent with the classical anatomical medial forebrain bundle (which is confluent with our imMFB) sensu stricto, but in our view—as a cortico–midbrain feedback structure—constitutes an integrative part of the much more complex human MFB [10]. Clinically, it appears conceivable to define TRD and OCD as network dysfunctions of the newly described MAINTENANCE system and there is good reason to understand dopamine as a key, albeit not singular, transmitter. As a consequence, the MFB as the anatomical substrate of the MAINTENANCE system is from a neuroethological viewpoint, a valid target for clinical interventions like DBS in these diseases for the future. 

## Figures and Tables

**Figure 1 brainsci-12-00438-f001:**
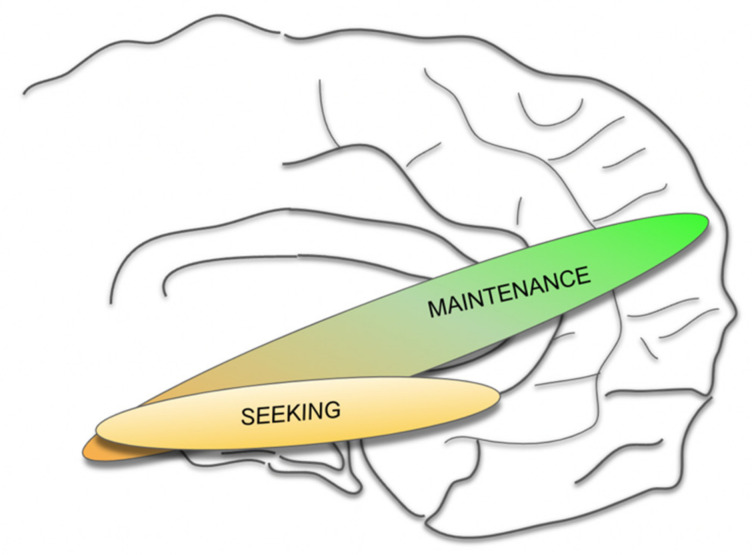
Proposed concept of integrated SEEKING and MAINTENANCE system.

**Figure 2 brainsci-12-00438-f002:**
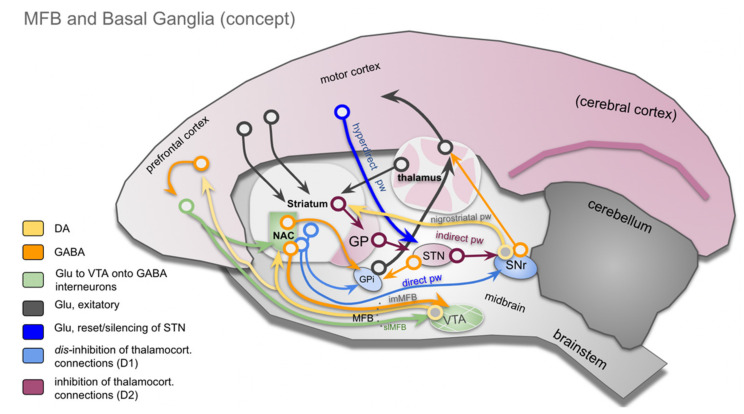
Basal ganglia integration of the MFB concept. *Direct, indirect and hyperdirect* (motor) *projections* are shown in parallel, as the basis for adaptive motor responses, allowing for flexible behavior. The nucleus accumbens (NAC) is part of the basal ganglia AND the limbic system. In this sense, the NAC is important for the motivational basis and regulation of flexible behaviors. Details in main text. (Legend: NAC, nucleus accumbens; GP, globus pallidus; GPi, globus pallidus internal segment; STN, subthalamic nucleus; SNr, substantia nigra pars reticulata; VTA, ventral tegmental area (of Tsai); MFB, medial forebrain bundle (primate); imMFB, inferomedial MFB; slMFB, superolateral MFB. (Adapted after [32] and arbitrarily set on marmoset background for visualization.).

**Figure 3 brainsci-12-00438-f003:**
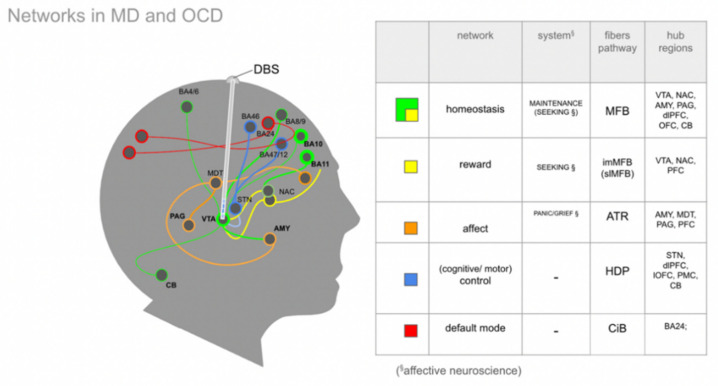
Overview of four networks relevant in major depressive disorder (MD), according to [53], and obsessive–compulsive disorder (OCD), extended by one overarching system for homeostasis, the MAINTENANCE system. Subcortical extensions according to [19]. A DBS electrode is positioned in the slMFB as part of the greater MFB system. Colored lines represent fiber connections; spheres represent network hubs. Capitalization of SEEKING and PANIC refers to affective neuroscience nomenclature [27] and to subcortical networks [19] which show affect-driven behavior upon stimulation. In analogy, the homeostasis network is termed MAINTENANCE since it reacts to direct stimulation (DBS) in a clinical setting. (Legend: AMY, amygdala; CB, cerebellum; CIB, cingulate bundle; DBS, Deep Brain Stimulation (lead); MDT, mediodorsal thalamus; NAC, nucleus accumbens; STN, subthalamic nucleus; VTA, ventral tegmental area; BA, Brodmann’s areas [54]; dlPFC, dorsolateral prefrontal cortex; OFC, orbitofrontal cortex; lOFC, lateral OFC; PFC, prefrontal cortex; PMC, premotor cortex; PAG, periaqueductal gray).

**Figure 4 brainsci-12-00438-f004:**
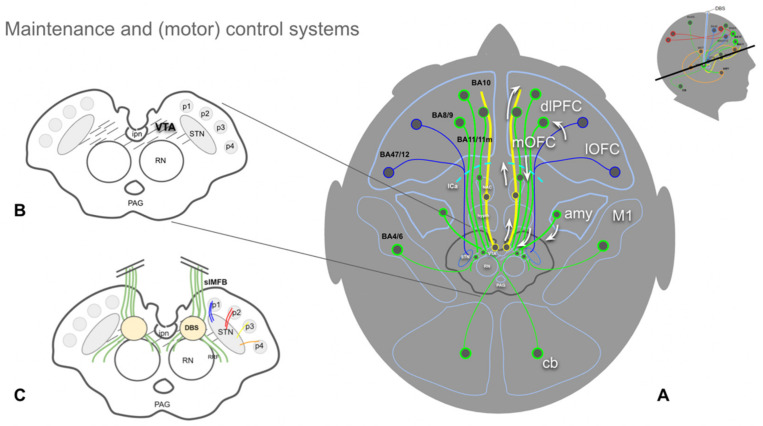
Overview of the MFB (MAINTENANCE) and part of the (motor) control system. (**A**), yellow fibers represent the imMFB and, therefore, the principal DA projection to cortical and subcortical structures. Green fibers represent cortical and cerebellar feedback projections to the VTA (Glu). Blue fibers visualize (part of) the motor control system, with projections from the lOFC and dlPFC to the STN (Glu, hyperdirect pathway). (**B**), midbrain topographic overview. The hatched region (VTA) is the principal origin of the DA fibers (A10 and medial A9 cell groups). (**C**), same as (**B**) but with slMFB fibers (green) included. Descending fibers diverge in front of the red nucleus (RN) to reach the VTA and raphe as well as the lateral tegmentum (RRF, retro-rubral field, DA group A8). (Legend: amy, amygdala; ipn, interpeduncular nucleus; DBS, deep brain stimulation electrode; STN, subthalamic nucleus; PAG, periaqueductal gray; p1, frontopontine tract; p2, corticobulbar tract; p3, corticospinal tract; p4; occipito-temporo-pontine tract; cb, cerebellum; OFC, orbitofrontal cortex; mOFC, medial and central OFC; lOFC, lateral OFC; dlPFC, dorsolateral prefrontal cortex; M1, primary motor cortex; BA, Brodmann’s area.)

**Figure 5 brainsci-12-00438-f005:**
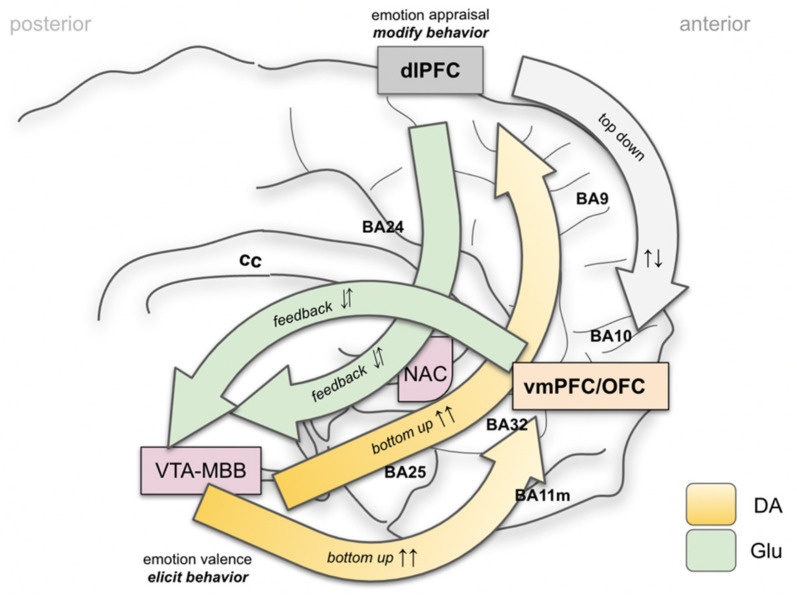
Simplified mechanisms of emotion perception and regulation of flexible behavior (most of basal ganglia and thalamus left out). Bottom-up (fast) and top-down (slow) mechanisms. The MFB system (close to VTA–MBB complex in conjunction with the episodic memory Papez’ circuit) realizes a primary interpretation of interoceptive and exteroceptive signals (affect valence) and rates these as *appetitive, aversive or neutral*. This system (MAINTENANCE) allows for engagement towards or away from a given stimulus. The bottom-up system informs simultaneously about the cognitive (to dlPFC) and emotional (vmPFC, OFC) consequences. The top-down system allows an appraisal of the salience of an emotional context. It is assumed to be less cost effective and slow. (Legend: vmPFC, ventromedial PFC; OFC, orbitofrontal cortex; cc, corpus callosum; BA, Brodmann’s area; NAC, nucleus accumbens).

**Figure 6 brainsci-12-00438-f006:**
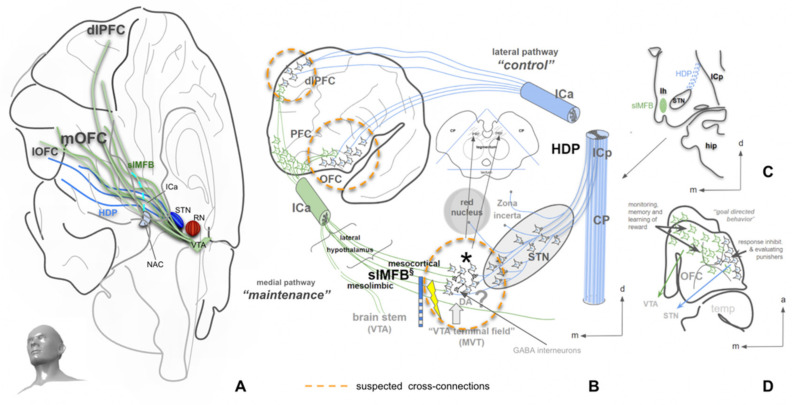
Topographic overview of midbrain long range connectivity. (**A**), view into the right hemisphere from anterior and inferior showing the coexisting hyperdirect pathway (blue fibers, control network) and the slMFB (green fibers, reward/MAINTENANCE network). (**B**), connection of lOFC and dlPFC via a lateral trans posterior capsule route to the subthalamic nucleus (blue, hyperdirect pathway). OFC to midbrain VTA trans anterior limb of internal capsule, trans-hypothalamic connection (green, slMFB). (**C**), coronal cut exemplifying lateral and medial entry of the distinct pathways. (**D**), view from inferior showing the origin and the overlap of distinctive pathways. (Legend: OFC, orbitofrontal cortex; lOFC, lateral OFC; mOFC, medial and central OFC; PFC, prefrontal cortex; dlPFC, dorsolateral prefrontal cortex; slMFB, superolateral medial forebrain bundle; HDP, hyperdirect pathway; NAC, nucleus accumbens; STN, subthalamic nucleus; RN, red nucleus; ICa, internal capsule anterior limb; ICp, internal capsule posterior limb; CP, cerebral peduncle; MVT, mesencephalic ventral tegmentum; hip, hippocampus; *, VTA terminal field of slMFB. B-D adapted after [10]).

**Figure 7 brainsci-12-00438-f007:**
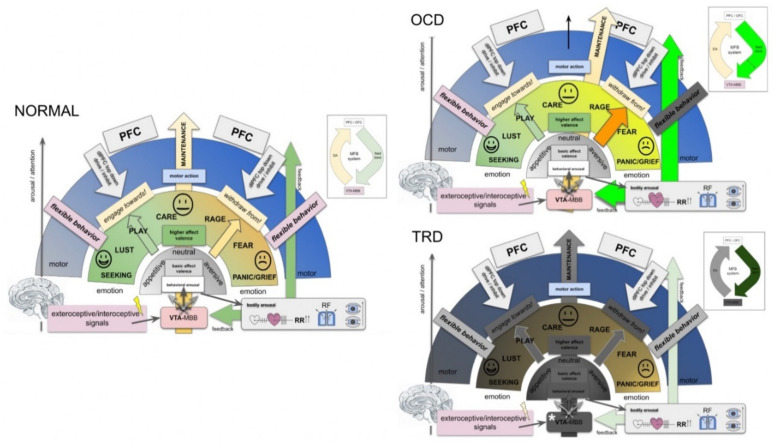
The MAINTENANCE system, a concept for the generation of flexible behavior and a reflection on pathological alterations in OCD and TRD. *NORMAL:* Via the dopaminergic midbrain cell groups (e.g., A8–A10; mostly VTA), a basal behavioral arousal is realized leading to a basic affect valence (appetitive <> aversive). The VTA is evolutionarily ancient and closely connected to Papez’ episodic memory circuitry [84] via the mammillary bodies (MBB). Behavioral arousal is accompanied by immediate bodily arousal to prepare the organism for a motor response. As such, this system inherently foreshadows such a motor response. Only after behavioral arousal and basic affect valence have occurred, higher affect valence and interpretation of interoceptive and exteroceptive signals takes place and basic emotions (here conceptualized as Panksepp’s seven emotional systems, [27]) are evaluated. These systems are realized via distinct and proprietary higher systems with anatomical architectural expression (and also as part of cortical networks). Higher affect valence leads to the initiation of a motor response. In a top-down mechanism, dlPFC can at any time control and moderate the execution of emotionally driven motor programs, resulting in flexible behavior. As such, emotionally driven and speedily initiated behavior remains under dynamic frontal lobe top-down control. OCD: Intrusive and ego-dystonic thoughts enter the system as an interoceptive signal (obsession). These thoughts are aversive and trigger meaningless motor programs (compulsions). A high level of emotionally induced motor activity allows only reduced flexible behavior with respect to obsessions. TRD: The MAINTENANCE system appears degraded. For details, see the text. Of note: Panksepp’s basic emotions are here not ranked according to their individual valence. Their occurrence in the image is arbitrary and only enlisted based on the authors’ personal preference. In this sense, RAGE is potentially not more or less aversive than PANIC, etc. Inset on upper right exemplifies the feedback mechanism. (Legend: RR, blood pressure; RF, respiratory frequency; MBB, mammillary bodies; VTA, ventral tegmental area; PFC, prefrontal cortex; dlPFC, dorsolateral prefrontal cortex).

## Data Availability

Not applicable.

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
