# Peer review of "“The Heart Asks Pleasure First”—Conceptualizing Psychiatric Diseases as *MAINTENANCE* Network Dysfunctions through Insights from slMFB DBS in Depression and Obsessive–Compulsive Disorder"

_brainsci, 2022, doi:10.3390/brainsci12040438_

Round 1

Reviewer 1 Report

Coenen and colleagues in the present perspective article entitled ‘ “The heart asks pleasure first“ - Conceptualizing psychiatric diseases as MAINTENANCE network circupathies through in sights from slMFB DBS in Depression and Obsessive-Compulsive Disorder’ , explored the current knowledge of the function and role of the medial forebrain bundle system (MFB) for reward and the integrated seeking/maintenance circuitry (which allows for the individual homeostasis and well-being), by integrating recent insights from studies which have applied deep brain stimulation over the superolateral medial forebrain bundle (slMFB) in treatment of medically therapy refractory depression (TRD) and obsessive-compulsive disorder (OCD).

The main strength of this manuscript is that it addresses an interesting and timely question, providing a captivating interpretation and describing how psychiatric diseases like OCD and TRD could be defined pathologies linked to this altered maintenance system. In general, I think the idea of this perspective article is really interesting and the authors’ fascinating observations on this timely topic may be of interest to the readers of Brain Sciences. However, some comments, as well as some crucial evidence that should be included to support the author’s argumentation, needed to be addressed to improve the quality of the manuscript, its adequacy, and its readability prior to the publication in the present form. My overall judgment is to publish this perspective article after the author have carefully considered my suggestions below, in particular reshaping parts of the Introduction and Discussion sections by adding more evidence.

Please consider the following comments:

  • Regarding the Abstract: according to the Journal’s guidelines, authors should have provided an abstract of about 200 words maximum. Indeed, the current one includes 268 words. Please correct it.
  • Keywords: Please consider adding ‘Therapy Refractory Depression (TRD)’ as a keyword.
  • I take issue with the authors’ frequent use of the term ‘circupathies’. I am not sure this is a proper word/definition, considering that there is no evidence of this word in any scientific or not database (i.e., Pubmed, Google Scholar, Researchgate, Google Servers). Please provide a clear and detailed explanation of the word’s etymology and/or semantics.
  • Introduction, line 58: specify the full name “Treatment Resistant Depression” and “Obsessive-Compulsive Disorder” before the acronyms “TRD” and “OCD”.
  • Brain wide networks relevant for Depression and OCD: The ‘Introduction’ section is well-written and nicely presented. Nevertheless, I believe that more detailed information about neural activity within the amygdala, medial prefrontal cortex as a core component of the anterior cingulate cortex, and the hippocampus, will provide a better and more accurate background when describing the emotional stimuli interpretation and regulation process. Thus, I suggest the authors to make anm effort to provide a brief overview of the pertinent published literature that offer a perspective on brain networks relevant for Depression and OCD and how they are crucial to interpret emotional valence to design flexible behavior, because as it stands, this information is not highlighted in the text. In particular, according with this sentence, I would suggest some crucial references that will methodologically fit with the present manuscript, for example a recent review (https://doi.org/10.1016/j.neubiorev.2021.04.036) on the ability of non-invasive brain simulation (NIBS) to interfere with activity of neural circuits (i.e., amygdala-mPFC-hippocampus) involved in the acquisition and consolidation of memories. Similarly, I would recommend another recent study (https://doi.org/10.1016/j.jad.2021.02.076) that illustrated the therapeutic potential of NIBS as a valid alternative in the treatment of abnormally persistent memories that characterized those patients with anxiety disorders that do not respond to psychotherapy and/or drug treatments, usually patients with an alteration of the amygdala-mPFC-hippocampus neural network. I believe that adding information regarding how the amygdala-mPFC-hippocampus brain network (i.e., emotional learning network) seems to be recruited in emotion perception and flexible behavior regulation will make a remarkable contribution to the topic's relevance.
  • The human medial forebrain bundle (MFB): In this section, authors described human medial forebrain bundle. In according with the previous comment, when they stated that ‘The imMFB is in our understanding essentially confluent with the mesocortico-limbic dopaminergic pathways and realizes as such the modulation of subcortical and cortical (PFC) structures mediating reward, reward learning as well as aversive signalling’, I believe that it may be useful to introduce some additional information about the role of frontal areas in reward learning. Thus, to have a more accurate representation of frontal lobe top-down control, I think that it could be useful to investigate the neurobiology of motivation and learning, by adding more information about neural circuitry that underlies the expression of motivated behavior. Specifically, I would suggest exploring the contribution that ventral prefrontal cortex (vmPFC) has to the encoding of internal and external factors controlling the perceived motivational value, highlighting how frontal lobe dysfunction affects individuals’ memory and emotional learning capacity, for example ‘revaluing the role of vmPFC in the acquisition of threat conditions’, and exploring ‘ventromedial prefrontal cortex subregional contributions in fear learning and fear extinction’.
  • Selected metal storage diseases and cognitive and motor rehabilitation: I suggest Authors to reorganize/rewrite this paragraph because, as it stands, this section is too thin and describes the research procedures in an excessively broad way.
  • Discussion: Considering that this is a perspective article, in my opinion, in this section the authors should have taken the opportunity to provide their interpretation of the implementation of deep brain stimulation in supero-lateral medial forebrain bundle to treat TRD and OCD, discuss the developments that are likely to be important in the future, and the avenues of research likely to become exciting as further studies yield more detailed results. Some questions the authors might consider and further explore in this section: How could the advances or research being discussed impact real-world outcomes (diagnosis, treatment guidelines, effectiveness, economics, drug utilization etc.)? Can changes be realistically implemented into clinical/research practice? What are the key areas for improvement in TRD and OCD treatment and how can current problems and limitations be solved?
  • In my opinion, I think the ‘Conclusions’ paragraph would benefit from some thoughts as well as in-depth considerations by the authors, because as it stands, it is very descriptive but not enough theoretical as a discussion should be. Authors should make an effort, trying to explain the theoretical implication as well as the translational application of their research.
  • Tables: Please consider including summary tables that could provide the readers with the information at one glance.
  • References: Authors should consider revising the bibliography, as there are several incorrect citations. Indeed, according to the Journal’s guidelines, they should provide the abbreviated journal name in italics, the year of publication in bold, the volume number in italics for all the references.
  • According to the Journal’s guidelines, authors should have included the ‘Institutional Review Board Statement’ and ‘Data Availability Statement’ sections under the “Back matter” heading.

Overall, this review contains 7 figures and 86 references. The manuscript might carry important value presenting how psychiatric diseases like OCD and TRD could be defined as pathologies of this altered maintenance system.

I hope that, after these careful revisions, the manuscript can meet the Journal’s high standards for publication.

I am available for a new round of revision of this perspective article.

Best regards,

Reviewer

Author Response

Coenen and colleagues in the present perspective article entitled ‘ “The heart asks pleasure first“ - Conceptualizing psychiatric diseases as MAINTENANCE network circupathies through in sights from slMFB DBS in Depression and Obsessive-Compulsive Disorder’ , explored the current knowledge of the function and role of the medial forebrain bundle system (MFB) for reward and the integrated seeking/maintenance circuitry (which allows for the individual homeostasis and well-being), by integrating recent insights from studies which have applied deep brain stimulation over the superolateral medial forebrain bundle (slMFB) in treatment of medically therapy refractory depression (TRD) and obsessive-compulsive disorder (OCD).

The main strength of this manuscript is that it addresses an interesting and timely question, providing a captivating interpretation and describing how psychiatric diseases like OCD and TRD could be defined pathologies linked to this altered maintenance system. In general, I think the idea of this perspective article is really interesting and the authors’ fascinating observations on this timely topic may be of interest to the readers of Brain Sciences. 

-> Thank you for this kind evaluation.

However, some comments, as well as some crucial evidence that should be included to support the author’s argumentation, needed to be addressed to improve the quality of the manuscript, its adequacy, and its readability prior to the publication in the present form. My overall judgment is to publish this perspective article after the author have carefully considered my suggestions below, in particular reshaping parts of the Introduction and Discussion sections by adding more evidence.

-> ok

Please consider the following comments:

  • Regarding the Abstract: according to the Journal’s guidelines, authors should have provided an abstract of about 200 words maximum. Indeed, the current one includes 268 words. Please correct it.

-> the abstract was significantly shortened and rewritten

  • Keywords: Please consider adding ‘Therapy Refractory Depression (TRD)’ as a keyword.

-> TRD as a keyword was added.

  • I take issue with the authors’ frequent use of the term ‘circupathies’. I am not sure this is a proper word/definition, considering that there is no evidence of this word in any scientific or not database (i.e., Pubmed, Google Scholar, Researchgate, Google Servers). Please provide a clear and detailed explanation of the word’s etymology and/or semantics.

-> this is a really good point. Thank you for pointing this out. We have used the term internally but more in a colloquial fashion. We have replaced  this term with the better expression  “network pathologies”

  • Introduction, line 58: specify the full name “Treatment Resistant Depression” and “Obsessive-Compulsive Disorder” before the acronyms “TRD” and “OCD”.
    -> was changed
  • Brain wide networks relevant for Depression and OCD: The ‘Introduction’ section is well-written and nicely presented. Nevertheless, I believe that more detailed information about neural activity within the amygdala, medial prefrontal cortex as a core component of the anterior cingulate cortex, and the hippocampus, will provide a better and more accurate background when describing the emotional stimuli interpretation and regulation process. Thus, I suggest the authors to make anm effort to provide a brief overview of the pertinent published literature that offer a perspective on brain networks relevant for Depression and OCD and how they are crucial to interpret emotional valence to design flexible behavior, because as it stands, this information is not highlighted in the text. In particular, according with this sentence, I would suggest some crucial references that will methodologically fit with the present manuscript, for example a recent review (https://doi.org/10.1016/j.neubiorev.2021.04.036) on the ability of non-invasive brain simulation (NIBS) to interfere with activity of neural circuits (i.e., amygdala-mPFC-hippocampus) involved in the acquisition and consolidation of memories. Similarly, I would recommend another recent study (https://doi.org/10.1016/j.jad.2021.02.076) that illustrated the therapeutic potential of NIBS as a valid alternative in the treatment of abnormally persistent memories that characterized those patients with anxiety disorders that do not respond to psychotherapy and/or drug treatments, usually patients with an alteration of the amygdala-mPFC-hippocampus neural network. I believe that adding information regarding how the amygdala-mPFC-hippocampus brain network (i.e., emotional learning network) seems to be recruited in emotion perception and flexible behavior regulation will make a remarkable contribution to the topic's relevance.

-> WE have extended our paragraph on networks and included the suggested papers.

  • The human medial forebrain bundle (MFB): In this section, authors described human medial forebrain bundle. In according with the previous comment, when they stated that ‘The imMFB is in our understanding essentially confluent with the mesocortico-limbic dopaminergic pathways and realizes as such the modulation of subcortical and cortical (PFC) structures mediating reward, reward learning as well as aversive signaling’, 

I believe that it may be useful to introduce some additional information about the role of frontal areas in reward learning. Thus, to have a more accurate representation of frontal lobe top-down control, I think that it could be useful to investigate the neurobiology of motivation and learning, by adding more information about neural circuitry that underlies the expression of motivated behavior. Specifically, I would suggest exploring the contribution that ventral prefrontal cortex (vmPFC) has to the encoding of internal and external factors controlling the perceived motivational value, highlighting how frontal lobe dysfunction affects individuals’ memory and emotional learning capacity, for example ‘revaluing the role of vmPFC in the acquisition of threat conditions’, and exploring ‘ventromedial prefrontal cortex subregional contributions in fear learning and fear extinction’.

-> We have included a paragraph on frontal lobe function with respect to the indicated topics. We have included this in the paragraph about “brain wide networks”. We have kept this brief since this is a discussion of its own and would be beyond the scope of our work. 

  • Selected metal storage diseases and cognitive and motor rehabilitation: I suggest Authors to reorganize/rewrite this paragraph because, as it stands, this section is too thin and describes the research procedures in an excessively broad way.

->  this must have slipped in from somewhere. There is no such paragraph in our paper.

  • Discussion: Considering that this is a perspective article, in my opinion, in this section the authors should have taken the opportunity to provide their interpretation of the implementation of deep brain stimulation in supero-lateral medial forebrain bundle to treat TRD and OCD, discuss the developments that are likely to be important in the future, and the avenues of research likely to become exciting as further studies yield more detailed results. Some questions the authors might consider and further explore in this section: How could the advances or research being discussed impact real-world outcomes (diagnosis, treatment guidelines, effectiveness, economics, drug utilization etc.)? Can changes be realistically implemented into clinical/research practice? What are the key areas for improvement in TRD and OCD treatment and how can current problems and limitations be solved?

-> we have introduced a paragraph on this topic but in order to not become too broad we have reduced it to the development of DBS as a treatment. We hope this is ok.

  • In my opinion, I think the ‘Conclusions’ paragraph would benefit from some thoughts as well as in-depth considerations by the authors, because as it stands, it is very descriptive but not enough theoretical as a discussion should be. Authors should make an effort, trying to explain the theoretical implication as well as the translational application of their research.
    -> The conclusion was in part rewritten
  • Tables: Please consider including summary tables that could provide the readers with the information at one glance.

-> We think that - as it is - the figures help the reader to understand the concept. We hope teh reviewer is ok with this decision. 

  • References: Authors should consider revising the bibliography, as there are several incorrect citations. Indeed, according to the Journal’s guidelines, they should provide the abbreviated journal name in italics, the year of publication in bold, the volume number in italics for all the references.

-> we have thoroughly revised the reference list. 

  • According to the Journal’s guidelines, authors should have included the ‘Institutional Review Board Statement’ and ‘Data Availability Statement’ sections under the “Back matter” heading.

-> I am not sure if this is intended as a heading. We have, however, changed this part according to the journal’s guidelines.

Overall, this review contains 7 figures and 86 references. The manuscript might carry important value presenting how psychiatric diseases like OCD and TRD could be defined as pathologies of this altered maintenance system.

I hope that, after these careful revisions, the manuscript can meet the Journal’s high standards for publication.

I am available for a new round of revision of this perspective article.

Best regards,

Reviewer

Reviewer 2 Report

The study aims to review the function and role of the MFB for reward and SEEKING by integrating recent insights from slMFB DBS in TRD and OCD. The authors formulate a re-interpretation of the MFB as the main structure of an integrated SEEKING/MAINTENANCE circuitry. In addition, the role of dopamine was emphasized in regulating flexible behavior. In general, this paper is interesting. The topic of research is innovative for understanding the function and role of the MFB and further extended role to the MFB which is the brain's homeostasis or MAINTENANCE system. The paper describes TRD and OCD as circupathies of the MAINTENANCE system and it conceive dopamine as a key albeit not singular transmitter.

Major concerns:

Point 1: The methods section describing review of previous literature is written with insufficient detail. 

Point 2: In addition, in methods section, the authors pointed out that they performed review related to VTA,DA, and slMFB,slMFB DBS. However. the Results section now only includes the observations from DBS surgery, please add the results or information about the others.

Point 3: Furthermore, the chief issue with the results is the lack of quantification. If other results could be obtained, then these should be added,  it would be more persuasive.

Point 4: The format of the reference document is informal. Please check these carefully.

Author Response

The study aims to review the function and role of the MFB for reward and SEEKING by integrating recent insights from slMFB DBS in TRD and OCD. The authors formulate a re-interpretation of the MFB as the main structure of an integrated SEEKING/MAINTENANCE circuitry. In addition, the role of dopamine was emphasized in regulating flexible behavior. In general, this paper is interesting. The topic of research is innovative for understanding the function and role of the MFB and further extended role to the MFB which is the brain's homeostasis or MAINTENANCE system. The paper describes TRD and OCD as circupathies of the MAINTENANCE system and it conceive dopamine as a key albeit not singular transmitter.

-> Thank you for this evaluation.

Major concerns:

Point 1: The methods section describing review of previous literature is written with insufficient detail. 

Point 2: In addition, in methods section, the authors pointed out that they performed review related to VTA,DA, and slMFB,slMFB DBS. However. the Results section now only includes the observations from DBS surgery, please add the results or information about the others.

-> we have taken care of this and added a short paragraph pointing towards the discussion.

Point 3: Furthermore, the chief issue with the results is the lack of quantification. If other results could be obtained, then these should be added,  it would be more persuasive.

-> We agree. However, since this is only a perspective paper we would like to stick to the anecdotal information included. Our previous work has focused on many of the quantitative aspects and was cited. At this moment we are conducting the FORESEE III trial (for TRD)  and can at this moment not go into further detailed evaluations.

Point 4: The format of the reference document is informal. Please check these carefully.

-> Reference list was checked.

Round 2

Reviewer 1 Report

I am very pleased to see that the authors have welcomed my suggestions and have clarified several of the questions I raised in my first round of this review. I believe that this article does an excellent work investigating how psychiatric diseases, like obsessive-compulsive disorder (OCD) or medically therapy refractory depression (TRD), could be defined as pathologies linked to alteration of the forebrain bundle system (MFB).

I only have one last minor suggestion to do: to further improve the theoretical background of the present article and its argumentation, I believe that it could be useful to add more information about neural circuitry that underlies the expression of motivated behavior. Specifically, I would suggest focusing on the contribution that the ventral prefrontal cortex (vmPFC) has to the encoding of internal and external factors controlling the perceived motivational and emotional value, therefore highlighting how frontal lobe dysfunction affects individuals’ memory and emotional learning capacity: evidence from a recent perspective manuscript (https://doi.org/10.17219/acem/146756) provided information for a deeper understanding of human learning neural networks, particularly on human PFC crucial role, that might also contribute to the advancement of alternative, more precise and individualized treatments for a variety of psychiatric disorders.

Overall, this is a timely and needed manuscript, and I look forward to seeing further study on this issue by these authors in the future.

Author Response

I am very pleased to see that the authors have welcomed my suggestions and have clarified several of the questions I raised in my first round of this review. I believe that this article does an excellent work investigating how psychiatric diseases, like obsessive-compulsive disorder (OCD) or medically therapy refractory depression (TRD), could be defined as pathologies linked to alteration of the forebrain bundle system (MFB).

-> Thank you.

I only have one last minor suggestion to do: to further improve the theoretical background of the present article and its argumentation, I believe that it could be useful to add more information about neural circuitry that underlies the expression of motivated behavior. Specifically, I would suggest focusing on the contribution that the ventral prefrontal cortex (vmPFC) has to the encoding of internal and external factors controlling the perceived motivational and emotional value, therefore highlighting how frontal lobe dysfunction affects individuals’ memory and emotional learning capacity: evidence from a recent perspective manuscript (https://doi.org/10.17219/acem/146756) provided information for a deeper understanding of human learning neural networks, particularly on human PFC crucial role, that might also contribute to the advancement of alternative, more precise and individualized treatments for a variety of psychiatric disorders.

-> We have added a small paragraph to this context.